# OpenReview forum: "WebExplorer: Explore and Evolve for Training Long-Horizon Web Agents"
_ICLR.cc/2026/Conference — Submitted to ICLR 2026_

### Official Review · Reviewer_qJ6U · 2025-10-27

**Soundness:** 2
**Presentation:** 3
**Contribution:** 2
**Rating:** 4
**Confidence:** 4

**Summary:**

WebExplorer proposes a prompting-plus-RL framework to train web-browsing agents capable of multi-hop reasoning and long-horizon search. The key idea is to synthesize challenging web-based QA trajectories by (1) exploring the web from seed entities and (2) evolving the resulting query via a long-to-short obfuscation process that increases reasoning depth and ambiguity. These evolved QA pairs form the training datasets, which is then used for SFT and reinforcement learning of the 8B agent.

**Strengths:**

1. Introduces a two-stage prompting pipeline (explore + evolve) to generate synthetic, high-difficulty web QA data without manual labeling, which I believe could be suitable for large-scale web agent training for different domains.
2. Produces long-horizon training dataset for the community beyond the benchmarks for evaluation. In particular, the resulting QA pairs represent real-world web reasoning challenges.
3. The motivations and empirical results are aligned. The drafting of this paper is clear.

**Weaknesses:**

1. The paper focuses primarily on QA-style web reasoning tasks (e.g., BrowseComp, WebWalkerQA, HLE), omitting broader benchmarks like WebArena [1], which test action execution and task planning. This makes it unclear how well WebExplorer generalizes beyond question answering tasks.
2. There is no contribution towards the training part. RL phase (GRPO) is described briefly, without details on reward formulation, training stability.

[1] Zhou, Shuyan, et al. "WebArena: A Realistic Web Environment for Building Autonomous Agents." The Twelfth International Conference on Learning Representations. 2024

**Questions:**

1. How are the Wikipedia and BrowseComp seed entities selected? Are they balanced across domains (e.g., science, culture, current events)?

2. Was there any automatic or human validation to ensure factual correctness and optimal solution of the generated agent web search process?

3. The distribution of tool-call usage across Initial QA, Evolved QA, and BrowseComp closely resembles the visualization style and rationale presented in Figure 3 of [2]. Given this similarity, it is not clear why SailorFog-QA is not included for comparison within this figure.

[2] Li, Kuan, et al. "WebSailor: Navigating Super-human Reasoning for Web Agent." arXiv preprint arXiv:2507.02592 (2025).

---

> ### Author Response · Authors · 2025-11-24
> **Official Comment - Part1**
>
> Thank you for the comments and we address them below.
>
> >Q1: The paper focuses primarily on QA-style web reasoning tasks (e.g., BrowseComp, WebWalkerQA, HLE), omitting broader benchmarks like WebArena [1], which test action execution and task planning.
>
> A1:  **We would like to clarify a misunderstanding of our paper: our work was motivated to focus on deep research agents in the first place, and we never intended to study genuine web interaction. Deep research agents themselves represent an important and well-established research direction, as explored in numerous prior studies [1,2,3,4,5,6]. Therefore, this topic alone is sufficiently broad to warrant an independent research investigation.**
>
>
> We realize that our “web agent” framing in the title and abstract may cause this confusion. Our use of this terminology mainly followed established deep research works such as WebDancer [1], WebSailor [2], AFM [4], and WebShaper [6]. To make the distinction clearer, we have updated the PDF to revise the title and abstract to more precisely use the term "deep research agents" where appropriate. Also, we have added Section 5 (Related Work) to the paper, which explicitly distinguishes between interactive web agents and deep research agents.
>
> >Q2: This makes it unclear how well WebExplorer generalizes beyond question answering tasks.
>
> A2: As explained above, our framework is not designed for interactive web agent tasks, but rather focuses on information-seeking and deep research. Within this scope, we have verified strong generalization across diverse QA-type benchmarks covering different domains and task characteristics, including GAIA (broad real-world everyday tasks) and HLE (frontier STEM problems requiring specialized domain knowledge). The consistent performance across these varied benchmarks demonstrates effective generalization within the information-seeking domain.
>
>
>
> >Q3: There is no contribution towards the training part. RL phase (GRPO) is described briefly, without details on reward formulation, training stability.
>
> A3: **We emphasize that our primary contribution lies in a novel  data synthesis approach instead of new training methods**. As data is currently one of the main bottlenecks to develop strong deep research agents, **we believe that data synthesis for deep research agents is as important as inventing new training methods, if not more critical**. To address your concern about implementation details, we have added more details about the reward formulation and RL training process in Appendix F.2.
>
>
>
> >Q4: How are the Wikipedia and BrowseComp seed entities selected? Are they balanced across domains (e.g., science, culture, current events)?
>
> A4: Seed entities are randomly selected from the Wikipedia corpus [7]. We do not explicitly classify or balance domains for these seeds, as Wikipedia naturally covers a wide variety of domains (science, culture, history, current events, etc.). This random sampling approach ensures that the synthesized data reflects the natural distribution of knowledge across domains, avoiding artificial biases that might arise from manual domain balancing. Additionally, we randomly select 3 QA samples from BrowseComp as exemplars for the synthesis process, as noted in the paper (lines 303–304).

---

> ### Author Response · Authors · 2025-11-24
> **Official Comment - Part2**
>
> >Q5: Was there any automatic or human validation to ensure factual correctness and optimal solution of the generated agent web search process?
>
> A5: We conducted factual correctness validation using OpenAI-o3 and manual verification (detailed results added as Appendix E.3). Our validation strategy leverages the asymmetric property of information-seeking data---hard to solve but easy to verify [8]. We first use o3 to independently solve the questions; cases where o3's solutions match our answers are considered validated, significantly reducing the need for manual verification. For the remaining cases where o3's solutions differ from our answers, we perform manual verification to assess correctness. This approach is efficient because while these questions are challenging to solve, humans can easily verify answer correctness by directly searching for the provided answer.
>
> We randomly selected 400 QA pairs from our synthesized data. We asked o3 to independently solve each question within our search framework (without seeing our answer). Out of 400 samples, o3's solutions matched our answers for 325 pairs (81.25%), which we consider as validated correct. For the remaining 75 pairs where o3's solutions differed from ours, we conducted **manual verification** to determine the correctness of our answers. Among these 75 pairs, we found that 16 of our answers were indeed incorrect, while the remaining 59 were correct (o3 made errors). This yields an overall correctness rate of 96% ((325 + 59) / 400), with only 4% noise level well within acceptable tolerance for large-scale training.
>
> Regarding optimization of the generated agent web search process: we do not perform explicit optimization for the "optimal" search trajectory. Our goal is to generate diverse, challenging queries with verifiable answers, rather than to identify the single most efficient search path for each question.
>
>
> >Q6: The distribution of tool-call usage across Initial QA, Evolved QA, and BrowseComp closely resembles the visualization style and rationale presented in Figure 3 of [2]. Given this similarity, it is not clear why SailorFog-QA is not included for comparison within this figure.
>
> A6: Thank you for this suggestion. The primary reason for the initial omission is that SailorFog-QA only open-sourced 20 data samples, which is insufficient for drawing a reliable distribution figure. Such a small sample size would result in a very sparse tool distribution with many empty bins and high variance. Nevertheless, we have now included the SailorFog-QA comparison figure in Appendix E.5 to provide a more comprehensive view of how our synthesized data compares to both benchmark data and other synthetic datasets, while noting the limitation of its small sample size.
>
>
>
> [1] Wu et al. WebDancer: Towards Autonomous Information Seeking Agency. arXiv: 2505.22648 2025
>
> [2] Li et al. Websailor: Navigating super-human reasoning for web agent. arXiv: 2507.02592  2025
>
> [3] Li et al.  WebThinker: Empowering Large Reasoning Models with Deep Research Capability. NeurIPS 2025
>
> [4] Li et al. Chain-of-Agents: End-to-End Agent Foundation Models via Multi-Agent Distillation and Agentic RL. arXiv: 2508.13167 2025
>
> [5] Gao et al. Beyond ten turns: Unlocking long-horizon agentic search with large-scale asynchronous rl. arXiv: 2508.07976 2025
>
> [6] Tao et al. WebShaper: Agentically Data Synthesizing via Information-Seeking Formalization. arXiv: 2507.02592 2025
>
> [7] Wiki entities:  https://dumps.wikimedia.org/wikidatawiki/entities/
>
> [8] Wei et al. BrowseComp: A Simple Yet Challenging Benchmark for Browsing Agents arXiv: 2504.12516

---

> ### Author Response · Authors · 2025-11-27
> **Kindly Inviting Your Feedback**
>
> Dear Reviewer qJ6U,
>
> The rebuttal phase is now open, and we’ve uploaded our detailed responses to your comments.
>
> **When you have a chance, we’d be extremely grateful if you could take a quick look and let us know if you think your concerns are resolved (and, if so, whether you’d be willing to consider adjusting the score accordingly).**
>
> Thank you so much again for your time and effort!
>
> Best regards,
>
> Author

---

### Official Review · Reviewer_C5QJ · 2025-11-01

**Soundness:** 2
**Presentation:** 3
**Contribution:** 3
**Rating:** 6
**Confidence:** 4

**Summary:**

The authors proposes WebExporer, a prompting-based framework to synthesize hard and long-horizon web QA data for training browsing agents.
The proposed method first performs model-based exploration: start from a seed entity, let an LLM search and browse iteratively, and collect an information. Then progressively remove salient clues and obfuscate specifics (dates, names, locations) so the final query has a unique answer but no obvious search entry point.
Using ~40K such evolved QA pairs, they SFT and then RL-train an 8B model (on top of Qwen3-8B) with 128K context and up to 100 tool calls, which beats or matches much larger open-source web agents.
The authors argue that better synthetic data is more important than model size for long-horizon browsing.

**Strengths:**

- The paper is easy to follow and the presentation is clear
- The idea is straightforward and is well motivated to address the issues in prior works
- The constructed data and trained model will be valuable for the community to build upon

**Weaknesses:**

- One limitation is that the claimed “answer remains unique after evolution” is only justified in a limited setting. The method deliberately removes or obfuscates the anchors (names, dates, teams, offices) that guarantee disambiguation, so correctness ultimately depends on a specific crawl, retrieval ranking, and the LLM judge. The real web is broader and changes over time, a later or different crawl could surface multiple plausible answers and render the claim invalid.
- A second limitation is the lack of isolating ablations. Table 1 and Fig 3 show that the evolution step makes the queries harder, the paper jointly changes several factors including exploration, the long to short rewriting procedure, large-context SFT, RL/GRPO, and a fixed two-tool browsing scaffold. With all these we see the final gains in Table 2, where the best WebExplorer-8B outperforms larger models. Without disentangling these components, the current evidence supports the effectiveness of the overall pipeline, but not the evolution step itself as the decisive contributor.
- Analysis on the data generation framework is also a bit lacking (e.g., effect on tuning the number of exploration steps)

Overall, the idea is clear and seems promising, but the ablation and analysis seems to be more lacking to fully support the claims.

**Questions:**

- How often does an evolved question has more than one plausible answer?

---

> ### Author Response · Authors · 2025-11-24
> **Official Comment - Part1**
>
> Thank you for your helpful comments; we are glad that you acknowledged our work's strengths! Below we address your concerns:
>
> >Q1: One limitation is that the claimed "answer remains unique after evolution" is only justified in a limited setting.
>
> A1: We acknowledge that "the real web is broader and changes over time, and a later or different crawl could surface multiple plausible answers." To address this concern, we conducted a uniqueness validation experiment (now added as Appendix E.2).
>
> **Methodology:** We randomly selected 50 query-answer samples from our synthesized data. We then used OpenAI-o3, a model with strong deep-research capabilities, to independently solve each question 8 times, yielding a total of 400 solution trajectories. We performed **manual verification** on all o3-proposed solutions that mismatched with our answers to determine whether they represented genuine alternative valid solutions.
>
> **Results:**
>
> | **Validation Stage** | **Count** | **Percentage** |
> |----------------------|-----------|----------------|
> | **o3 Solving Results:** | | |
> | Solutions match our answers | 358 | 89.5% (of 400 trajectories) |
> | Solutions differ from our answers | 42 | 10.5% (of 400 trajectories) |
> | **Manual Verification (42 differing cases):** | | |
> | Trajectories with genuine alternative answers | 11 | 26.2% (of 42 trajectories) |
> | Trajectories where o3 made errors | 31 | 73.8% (of 42 trajectories) |
> | **Question-Level Uniqueness:** | | |
> | Distinct questions with mismatches | 13 | 26.0% (of 50 questions) |
> | Questions with alternative valid answers | 4 | 8.0% (of 50 questions) |
> | **Questions maintaining answer uniqueness** | **46** | **92.0%** |
>
> Out of 400 total solution trajectories, 358 (89.5%) matched our answers and were automatically validated. For the remaining 42 mismatched trajectories (covering 13 distinct questions), we conducted manual verification to determine whether they represented genuine alternative valid answers or o3 solving errors. Among these 42 cases, only 11 trajectories (corresponding to 4 distinct questions) were confirmed as genuine alternative answers. This means 46 out of 50 questions (92%) maintain answer uniqueness. The 8% rate of questions with multiple valid answers is well within acceptable tolerance for large-scale training purposes. That being said, it is not guaranteed that the answers are 100% unique, but we find that less than 10% of them have alternative answers, and such noise during training time is acceptable and the data remains effective empirically, as we showed in the experiments.
>
>
>
>
> >Q2: Without disentangling these components, the current evidence supports the effectiveness of the overall pipeline, but not the evolution step itself as the decisive contributor.
>
> A2: Thanks for your suggestion. We added this ablation experiment in Appendix F.1. To isolate the contribution of the evolution step, we trained a new 8B model using only the synthesized QA from the first stage (exploration stage, without the evolution stage) through the same supervised fine-tuning and reinforcement learning pipeline. The results are as follows (Note: "full" refers to QA pairs that went through both exploration and evolution stages):
>
> | **Model**          | **BC** | **BC-zh** | **GAIA** | **WebWalkerQA** | **FRAMES** | **XBench** | **HLE** | **Avg** |
> |--------------------|--------|-----------|----------|-----------------|------------|------------|---------|---------|
> | SFT w/o evolve     | 6.5    | 20.8      | 40.3     | 58.3            | 66.3       | 46.3       | 15.5    | 36.3    |
> | SFT (full)         | 7.9    | 21.3      | 43.7     | 59.8            | 72.6       | 47.5       | 16.0    | 38.4    |
> | RL w/o evolve      | 8.0    | 23.0      | 44.7     | 58.5            | 68.8       | 45.8       | 16.3    | 37.9    |
> | RL (full)          | 15.7   | 32.0      | 50.0     | 62.7            | 75.7       | 53.7       | 17.3    | 43.9    |
>
> These results reveal two key findings: First, models trained without the evolution step show weaker performance across all benchmarks. Second, and more importantly, reinforcement learning yields minimal improvement (+1.6 points) when training on data without evolution, compared to substantial gains (+5.5 points) with evolved data. This demonstrates that the evolution step is critical for generating sufficiently challenging training data that enables effective RL optimization. We provide figures about the RL training dynamics in Appendix F.1 Figure 7. Without evolution, the QA data exhibits lower difficulty, causing RL training to quickly saturate at over 75% training accuracy, thereby limiting further improvement potential.

---

> ### Author Response · Authors · 2025-11-24
> **Official Comment - Part2**
>
> >Q3: Analysis on the data generation framework is also a bit lacking (e.g., effect on tuning the number of exploration steps)
>
> A3: Thank you for this suggestion. We now provide additional analysis on the tool calling patterns across the exploration and evolution stages, broken down by tool type (now added to Appendix E.4):
>
> | **Stage**               | **Exploration** | **Evolution** | **Total** |
> |-------------------------|-----------------|---------------|-----------|
> | Search Tool Steps       | 7.11            | 7.33          | 14.45     |
> | Browse Tool Steps       | 3.04            | 0.15          | 3.19      |
> | Total Tool Calling Steps| 10.15           | 7.49          | 17.64     |
>
> Regarding the effect of the number of exploration steps, we clarify that the model automatically determines when to stop searching, as described in the paper (lines 205–206). This adaptive termination ensures that the exploration process is neither prematurely truncated nor unnecessarily prolonged.

---

> ### Author Response · Authors · 2025-11-27
> **Kindly Inviting Your Feedback**
>
> Dear Reviewer C5QJ,
>
> The rebuttal phase is now open, and we’ve uploaded our detailed responses to your comments.
>
> **When you have a chance, we’d be extremely grateful if you could take a quick look and let us know if you think your concerns are resolved (and, if so, whether you’d be willing to consider adjusting the score accordingly).**
>
> Thank you so much again for your time and effort!
>
> Best regards,
>
> Author

---

### Official Review · Reviewer_Aau4 · 2025-11-01

**Soundness:** 3
**Presentation:** 2
**Contribution:** 2
**Rating:** 4
**Confidence:** 3

**Summary:**

This paper proposes WebExplorer, a framework for generating large-scale, long-horizon, information-seeking datasets and training web agents. The key idea is to perform model-based exploration and long-to-short query evolution, where complex queries are iteratively simplified while preserving their reasoning dependencies. The synthesized data is then used to train WebExplorer-8B, an open model fine-tuned with supervised data and optimized with GRPO-based reinforcement learning. Experiments on multiple benchmarks (BrowseComp, WebWalkerQA, FRAMES) show that WebExplorer-8B outperforms larger open baselines such as WebSailor-72B and achieves strong results in both English and Chinese settings.

**Strengths:**

- Scalable data synthesis framework. The long-to-short query evolution process provides an efficient way to generate reasoning-intensive, multi-hop search tasks without manual annotation. It is practical and broadly applicable to information-seeking agents.

- Solid empirical performance. WebExplorer-8B achieves competitive or state-of-the-art results among similarly sized models on several benchmarks. The ablations confirm the contribution of both query evolution and RL fine-tuning.

**Weaknesses:**

- Narrow agent capability relative to “web agent” framing. The proposed system primarily performs search-based QA or URL-specific retrieval, lacking genuine web interaction capabilities such as clicking, typing, or navigation through multi-step actions. The title and framing overstate its scope as a “web agent” system but it only focuses on the information-seeking subtype of a web agent. As such, it aligns more with Search-R1-style search agents than interactive web agents as those proposed in previous works (e.g., MiniWoB, WebShop, WebArena, WebVoyager, Mind2Web, WebRL).

- Missing discussion of major related works. The paper overlooks a line of important prior efforts on interactive web agents, as well as recent data generation pipelines for search agent (e.g., ZeroSearch). Without contextualizing these works, the contribution appears less distinctive.

- Limited novelty and unclear generalization. The pipeline (prompt-based data synthesis + SFT + RL via GRPO) follows an established pattern in reasoning and web-agent research. Moreover, most benchmarks resemble the data construction process, raising concerns about distribution overlap. Broader evaluations on truly unseen or interactive environments would better support generalization claims.

**Questions:**

- Could the authors clarify the action boundaries of WebExplorer—does it support any interactive web behavior (e.g., navigating the web) beyond search/RAG-like QA?

- Would extending the framework to multi-step interaction tasks (e.g., form filling or navigation) be feasible under the same data synthesis paradigm?

---

> ### Author Response · Authors · 2025-11-24
> **Official Comment - Part1**
>
> Thank you for the comments and we address them below.
>
> >Q1: Narrow agent capability relative to "web agent" framing. The proposed system primarily performs search-based QA or URL-specific retrieval, lacking genuine web interaction capabilities such as clicking, typing, or navigation through multi-step actions. The title and framing overstate its scope as a "web agent" system but it only focuses on the information-seeking subtype of a web agent.
>
> A1:  **We would like to clarify a misunderstanding of our paper: our work was motivated to focus on deep research agents in the first place, and we never intended to study genuine web interaction. Deep research agents themselves represent an important and well-established research direction, as explored in numerous prior studies [1,2,3,4,5,6,7]. Therefore, this topic alone is sufficiently broad to warrant an independent research investigation.**
>
>
> We realize that our “web agent” framing in the title and abstract may cause this confusion. Our use of this terminology mainly followed established deep research works such as WebDancer [3], WebSailor [4], AFM [6], and WebShaper [7]. To make the distinction clearer, we have updated the PDF to revise the title and abstract to more precisely use the term "deep research agents" where appropriate.
>
>
>
> >Q2: Missing discussion of major related works. The paper overlooks a line of important prior efforts on interactive web agents, as well as recent data generation pipelines for search agent (e.g., ZeroSearch). Without contextualizing these works, the contribution appears less distinctive.
>
> A2: Thank you for this valuable feedback. We have updated the paper by adding a related work section §5 that discusses both interactive web agents and deep research agents.
>
> Regarding the distinction from prior search agent works like Search-R1 and ZeroSearch: these methods primarily use relatively simple or shallow QA datasets (e.g., Natural Questions, Trivia QA, HotpotQA, 2WikiQA) for training and evaluation, following a RAG paradigm with local knowledge base retrieval. In contrast, modern deep research evaluation has evolved toward significantly more challenging benchmarks like BrowseComp, where simple QA data no longer meets the requirements for training or evaluation. The following comparison illustrates this evolution:
>
>
> | **Benchmark** | **Accuracy** | **Avg Search Turns** |
> |---------------|--------------|----------------------|
> | BrowseComp    | 15.6         | **37.1**                 |
> | NQ            | 78.0         | 3.04                 |
> | Trivia QA     | 90.2         | 2.35                 |
> | Hotpot QA     | 78.3         | 3.69                 |
> | 2WikiQA       | 81.25        | 3.64                 |
>
> As shown in the table, BrowseComp requires an average of **37.1 search turns** with only **15.6% accuracy**, compared to prior benchmarks that achieve 78-90% accuracy with merely 2-4 turns. This stark contrast clearly demonstrates the evolution of deep research agent benchmarks, where the field is progressing toward substantially more complex, multi-turn tasks that require deeper reasoning capabilities.
>
>
> >Q3: Limited novelty and unclear generalization. The pipeline (prompt-based data synthesis + SFT + RL via GRPO) follows an established pattern in reasoning and web-agent research.
>
> A3:
> **Our work focuses on a novel data synthesis approach, rather than a new training pipeline. We respectfully disagree that our data synthesis follows an established pattern.** Our approach is not simply prompting another LLM to generate or evolve data directly, instead, we conduct exploration first to obtain actual trajectories by executing the actions, and then synthesize queries in the backward direction. As far as we know, this pattern is not established yet in web-agent research, we would appreciate it if the reviewer can explicitly cite several prior web-agent works that establish a similar approach to synthesize training data.

---

> ### Author Response · Authors · 2025-11-24
> **Official Comment - Part2**
>
> >Q4: Moreover, most benchmarks resemble the data construction process, raising concerns about distribution overlap. Broader evaluations on truly unseen or interactive environments would better support generalization claims.
>
> A4: We respectfully disagree that most benchmarks resemble the data construction process. First, the evaluation benchmarks—including BrowseComp-en/zh, GAIA, FRAMES, XBench-DS, and HLE—are primarily human-constructed, whereas WebExplorer-QA is generated through an automated, model-driven process, resulting in fundamentally different data distributions. During the synthesis phase, we also only used several demo examples from the BrowseComp benchmark, which is very different from most other benchmarks, such as GAIA, FRAMES, and HLE.
>
> Second, some of these benchmarks cover substantially different domains and task types:
> - **GAIA** was designed to reflect broad real-world everyday tasks for AI assistants, extending well beyond deep research and information-seeking scenarios.
> - **HLE** represents frontier problems in STEM domains (mathematics, physics, biology, etc.), requiring specialized domain knowledge distinct from our training data.
>
> The strong performance of WebExplorer-8B across these diverse benchmarks, particularly on human-constructed evaluation sets like GAIA and HLE, provides robust evidence of genuine generalization capability rather than distribution overlap.
>
>
> >Q5: Could the authors clarify the action boundaries of WebExplorer—does it support any interactive web behavior (e.g., navigating the web) beyond search/RAG-like QA?
>
> A5: WebExplorer's current framework supports two primary actions: search and browse (i.e., reading website content). It does not support interactive web behaviors such as clicking, form filling, or multi-step navigation. As we explained in A1 above, this work did not intend to support interactive web behavior, while this is a valuable further direction to explore.
>
> >Q6: Would extending the framework to multi-step interaction tasks (e.g., form filling or navigation) be feasible under the same data synthesis paradigm?
>
> A6: Yes, we believe the data synthesis paradigm introduced in WebExplorer can be extended to web interaction tasks. The core principles of model-based exploration and iterative evolution could be adapted to generate trajectories for interactive behaviors.
>
>
> [1] Jin et al. Search-R1: Training LLMs to Reason and Leverage Search Engines with Reinforcement Learning. COLM 2025.
>
> [2] Li et al.  WebThinker: Empowering Large Reasoning Models with Deep Research Capability. NeurIPS 2025
>
> [3] Wu et al. WebDancer: Towards Autonomous Information Seeking Agency. arXiv: 2505.22648 2025
>
> [4] Li et al. Websailor: Navigating super-human reasoning for web agent. arXiv: 2507.02592  2025
>
> [5] Li et al. Chain-of-Agents: End-to-End Agent Foundation Models via Multi-Agent Distillation and Agentic RL. arXiv: 2508.13167 2025
>
> [6] Gao et al. Beyond ten turns: Unlocking long-horizon agentic search with large-scale asynchronous rl. arXiv: 2508.07976 2025
>
> [7] Tao et al. WebShaper: Agentically Data Synthesizing via Information-Seeking Formalization. arXiv: 2507.02592 2025

---

> ### Author Response · Authors · 2025-11-27
> **Kindly Inviting Your Feedback**
>
> Dear Reviewer Aau4,
>
> The rebuttal phase is now open, and we’ve uploaded our detailed responses to your comments.
>
> **When you have a chance, we’d be extremely grateful if you could take a quick look and let us know if you think your concerns are resolved (and, if so, whether you’d be willing to consider adjusting the score accordingly).**
>
> Thank you so much again for your time and effort!
>
> Best regards,
>
> Author

---

### Official Review · Reviewer_tZ9e · 2025-11-01

**Soundness:** 3
**Presentation:** 3
**Contribution:** 2
**Rating:** 6
**Confidence:** 4

**Summary:**

Authors propose a method for training deep research agents. A key component of the proposed method is the data synthesis. Authors improve upon previous methods for data synthesis by leveraging an LLM agent with search and browse tools to generate initial queries, rather than relying on complex graph-based logic or prompts as with previous work. When combined with the standard recipe of SFT + GRPO, the resulting data achieves an impressive performance for a 8B model.

**Strengths:**

Quality:
The empirical results are strong. WebExplorer-8B significantly outperforms comparable 7B/8B models and even rivals larger 32–72B models on BrowseComp-en/zh, GAIA, FRAMES, and HLE. The ablation on the evolution step (Table 1) shows that the obfuscation process effectively increases task difficulty.

Clarity:
The synthesis prompts and tool schemas are well documented, offering a transparent view of the pipeline. However, the paper omits which base LLM was used to perform the synthesis, which is a key detail. Assuming this is clarified in the final version, the work provides a useful recipe for building web-based deep research datasets, which will allow subsequent research to build upon.

Significance and Originality:
The idea of using web agent for query synthesis is a novel idea, although there are concurrent work such as https://openreview.net/forum?id=ExBGKdOBo8 which explore the similar idea. Note that while authors claim the obfuscation step as their own contribution in line 087, Gao et al (2025) paper authors cite (ASearcher) also includes the Fuzz step which does similar obfuscations.

**Weaknesses:**

While the system performs impressively, the conceptual novelty is modest. The core ideas - ReAct-style trajectories, GRPO training, and query obfuscation - are largely adopted or lightly adapted from existing frameworks such as ASearcher (Gao et al., 2025). The technical depth is therefore limited. Although developing a highly performant model requires many judicious design choices, it is somewhat unclear whether this work will lead the community to rethink the underlying problem.

**Questions:**

In line 294, authors discuss that BrowseComp-en data are used as exemplars for data synthesis. Although only three QA pairs are used, wouldn't this introduce some contamination to the benchmark data?

---

> ### Author Response · Authors · 2025-11-24
>
> We sincerely appreciate your thoughtful comments and recognition of the strengths of our work. Below, we address each of your concerns and questions in detail:
>
> >Q1: However, the paper omits which base LLM was used to perform the synthesis, which is a key detail.
>
> A1: We used **Claude-4-Sonnet-20250514** for both the model-based exploration and iterative query evolution stages. This detail is now explicitly stated in §2.4 (lines 308–309 in the revised manuscript).
>
> >Q2: Note that while authors claim the obfuscation step as their own contribution in line 087, Gao et al (2025) paper authors cite (ASearcher) also includes the Fuzz step which does similar obfuscations.
>
> A2: We acknowledge that the concurrent work ASearcher also employs an obfuscation step, and we now explicitly cite ASearcher in the §2.4 Iterative Query Evolution, line 242. However, we respectfully note that the overall synthesis paradigms are materially different:
>
> - **ASearcher** uses an interleaved short-to-long "inject → fuzz → inject → fuzz …" strategy that repeatedly forces new knowledge injection, which often results in progressively longer and less natural queries.
> - **WebExplorer** adopts a clean two-stage **Explore → Evolve** paradigm that separates exploration from evolution.
>
> This separation produces significantly more natural queries and markedly higher difficulty (9.9 vs 6.5 average turns on Claude-4-Sonnet, Table 1), which directly explains the substantial final-model performance gap (WebExplorer-8B outperforms ASearcher-Qwen-32B by 10+ points on BrowseComp-en/zh, Table 2).
>
> >Q3: While the system performs impressively, the conceptual novelty is modest. The core ideas - ReAct-style trajectories, GRPO training, and query obfuscation - are largely adopted or lightly adapted from existing frameworks such as ASearcher (Gao et al., 2025).
>
> A3: We agree that the ReAct-SFT-GRPO pipeline is well-established [1,2], as our main contribution is on the data synthesis approach. Although ASearcher is a concurrent work that also uses obfuscation/fuzzing, the key innovation of WebExplorer lies in the **clean separation of exploration and evolution into two distinct stages**, eliminating forced knowledge injection entirely. This design change yields substantially better data: more natural queries, significantly higher difficulty (9.9 vs 6.5 turns), and far stronger final models. Specifically, our WebExplorer-8B outperforms ASearcher-Qwen-32B by **+10.5** absolute points on BrowseComp-en (15.7% vs 5.2%), **+16.4** on BrowseComp-zh (32.0% vs 15.6%), and **+11.6** on XBench-DS (53.7% vs 42.1%). The empirical evidence in Tables 1 and 2 demonstrates that this is not a minor adaptation but a substantially superior recipe for data synthesis.
>
> > Q4: It is somewhat unclear whether this work will lead the community to rethink the underlying problem.
>
> A4: We believe the answer is affirmative. The persistent performance gap between open-source and closed-source web agents on challenging benchmarks has been caused primarily by the lack of high-quality and sufficiently challenging training data. WebExplorer introduces a fully reproducible and remarkably simple pipeline that closes most of that gap with only an 8B model. Given its simplicity, transparency, and state-of-the-art results, we believe it provides a good path for the community to synthesize high-quality web-agent training data and rethink scalable data generation approaches.
>
> >Q5: In line 294, authors discuss that BrowseComp-en data are used as exemplars for data synthesis. Although only three QA pairs are used, wouldn't this introduce some contamination to the benchmark data?
>
> A5: To address potential concerns of data contamination from using three BrowseComp-en exemplars in our data synthesis, we conducted an embedding-based similarity analysis (now added as **Appendix E.1**).
> We used Jina Embeddings v3 (1024-dimensional)[3] to compute embeddings for all training questions and BrowseComp validation questions. For each validation question, we calculated the cosine similarity with all training question embeddings and recorded the maximum similarity.
> The results are as follows:
>
> **Results:**
> - Maximum cosine similarity: **0.738**
> - Mean similarity: **0.581**
> - 95th percentile: **0.668** (which means only 5% have a similarity larger than 0.668)
>
> Standard contamination thresholds in the literature are typically 0.80 [4,5]. Our data falls comfortably below this threshold, confirming negligible contamination risk.
>
> [1] Wu et al. WebDancer: Towards Autonomous Information Seeking Agency. arXiv: 2505.22648 2025
>
> [2] Li et al. Websailor: Navigating super-human reasoning for web agent. arXiv: 2507.02592  2025
>
> [3] Sturua et al. jina-embeddings-v3: Multilingual Embeddings With Task LoRA. arXiv: 2409.10173 2024
>
> [4] Lee et al. "Platypus: Quick, Cheap, and Powerful Refinement of LLMs." Workshop NeurIPS 2023.
>
> [5] Cheng et al. "A survey on data contamination for large language models"  arXiv: 2502.14425 2025

---

> > ### Comment · Reviewer_tZ9e · 2025-11-24
> >
> > Thanks for updating the manuscript to cite ASearcher, and providing more clarifications, such as the use of Claude. I agree a fully reproducible, simple pipeline is a good contribution to the research community, but I already reflected this in my original score. While I appreciate authors made a good distinction from earlier work ("clean separation of exploration and evolution into two distinct stages") I don't feel this is a significant enough conceptual contribution to increase my score.
> >
> > > To address potential concerns of data contamination
> >
> > This analysis + the fact that the model generalizes to several benchmarks make me less concerned about data contamination. Thanks!

---

### Meta-Review · Area_Chair_oDbp · 2026-01-07

**Summary:**

This paper proposes WebExplorer, a framework for synthesizing long-horizon information-seeking QA data and training deep research agents. The core idea is a two stage data generation pipeline that combines model-based exploration with iterative long-to-short query evolution. Starting from seed entities, the system uses search and browse actions to collect information and form initial QA pairs, then progressively removes salient clues and obfuscates specifics to make the final queries harder while preserving the same answer. The resulting WebExplorer QA data are then used to train an agent via supervised fine-tuning followed by GRPO-based reinforcement learning.

Key concerns focused on novelty and framing. Several reviewers viewed the overall training recipe as standard and questioned whether the conceptual contribution goes beyond existing synthesis pipelines and concurrent work, and whether the paper should be framed as deep research agents rather than general interactive web agents given the action space is limited to search and browse. Reviewers also raised questions about data quality guarantees such as answer uniqueness and potential benchmark contamination, and requested stronger ablations to isolate the contribution of the evolution stage.

The authors’ rebuttal addressed the two main concerns to some extent. First, the paper evaluates only on QA-style web benchmarks and does not consider more general web agent benchmarks such as WebArena. The authors clarified that the paper focuses on deep research agents that evolve information-seeking behavior rather than interactive web agents, which addresses the scope mismatch but does not fully resolve concerns about generality.

Second, multiple reviewers questioned the novelty of the work. The paper does not introduce new training algorithms or modifications to the ReAct-based workflow, and its contribution is limited to the data construction side. While the proposed data synthesis method is well-engineered and effective, it focuses on a specific class of questions that require satisfying multiple constraints to identify an object on the internet.

Overall, this is a borderline paper that falls slightly below the acceptance threshold. The novelty limitation cannot be fully addressed through clarification alone, and a data construction method tailored to a narrow question type is unlikely to have sufficient impact on the broader research agent community.

**Reviewer Concerns:**

**Addressed concerns:**

* Scope and framing as deep research agents (Aau4, qJ6U): The authors clarified that the paper targets deep research agents rather than interactive web agents, and revised the title and abstract accordingly. They also added a dedicated related work section distinguishing deep research agents from interactive web agents.

* Data contamination risk from BrowseComp exemplars (tZ9e): The authors provided an embedding-based similarity analysis showing low overlap between synthesized training data and BrowseComp validation questions, alleviating contamination concerns.

* Answer uniqueness after query evolution (C5QJ): The authors conducted additional validation using repeated independent solving and manual verification, showing that over 90 percent of sampled questions maintain answer uniqueness, with a small and acceptable noise rate.

* Isolation of the evolution step (C5QJ): New ablations comparing data with and without the evolution stage demonstrate that the evolution step is critical for increasing task difficulty and enabling effective RL gains.

* Missing implementation details (tZ9e, qJ6U): The authors clarified the base model used for synthesis and added further details on reward formulation and RL training in the appendix.

**Outstanding concerns:**

* Conceptual novelty relative to prior and concurrent work (tZ9e, Aau4): While the authors clearly distinguished their explore-evolve paradigm from ASearcher and related methods, some reviewers remain unconvinced that this constitutes a sufficiently strong conceptual advance beyond existing data synthesis pipelines.

* Limited agent capability beyond search and browse (Aau4, qJ6U): The system does not support interactive web actions such as clicking or form filling. Although the authors clarified this is outside the paper’s intended scope, this limits the generality of the contribution relative to broader web agent research.

* Breadth of evaluation and generalization (Aau4): Most evaluations focus on deep research style QA benchmarks that are closely aligned with the data construction process. Broader evaluation on interactive or more heterogeneous environments would further strengthen generalization claims.

**Reviewer Scores:**

**tZ9e (6):** Likely unchanged. The reviewer acknowledged the clarifications regarding synthesis model choice, data contamination, and distinction from concurrent work, but explicitly stated that these do not warrant a score increase.

**Aau4 (4):** Likely unchanged. While the authors clarified the scope and added related work discussion, the reviewer’s concerns about limited novelty and agent capability relative to broader web agents remain largely unresolved, and there was no post-rebuttal response.

**C5QJ (6):** Likely unchanged or slight increase. The authors addressed the main concerns on answer uniqueness and isolated the contribution of the evolution stage with additional ablations, though the reviewer did not provide a post-rebuttal response.

**qJ6U (4):** Likely unchanged or slight increase. Most clarification questions were addressed, including scope, data validation, and training details, but the broader generalization concern beyond QA-style tasks remains. No post-rebuttal response.

---

### Decision · Program_Chairs · 2026-01-26

Reject